# Low-Temperature-Meltable Elastomers Based on Linear Polydimethylsiloxane Chains Alpha, Omega-Terminated with Mesogenic Groups as Physical Crosslinker: A Passive Smart Material with Potential as Viscoelastic Coupling. Part II—Viscoelastic and Rheological Properties

**DOI:** 10.3390/polym12122840

**Published:** 2020-11-29

**Authors:** Sabina Horodecka, Adam Strachota, Beata Mossety-Leszczak, Maciej Kisiel, Beata Strachota, Miroslav Šlouf

**Affiliations:** 1Institute of Macromolecular Chemistry, Czech Academy of Sciences, Heyrovskeho nam. 2, CZ-162 06 Praha, Czech Republic; horodecka@imc.cas.cz (S.H.); beata@imc.cas.cz (B.S.); slouf@imc.cas.cz (M.Š.); 2Faculty of Science, Charles University, Albertov 6, CZ-128 00 Praha, Czech Republic; 3Faculty of Chemistry, Rzeszow University of Technology, al. Powstancow Warszawy 6, PL-35-959 Rzeszow, Poland; mossety@prz.edu.pl (B.M.-L.); m.kisiel@prz.edu.pl (M.K.)

**Keywords:** reversible networks, self-assembly, self-healing, liquid crystals, smart materials, rheology

## Abstract

Rheological and viscoelastic properties of physically crosslinked low-temperature elastomers were studied. The supramolecularly assembling copolymers consist of linear polydimethylsiloxane (PDMS) elastic chains terminated on both ends with mesogenic building blocks (LC) of azobenzene type. They are generally and also structurally highly different from the well-studied LC polymer networks or LC elastomers: The LC units make up only a small volume fraction in our materials and act as fairly efficient physical crosslinkers with thermotropic properties. The aggregation (nano-phase separation) of the relatively rare, small and spatially separated terminal LC units generates temperature-switched viscoelasticity in the molten copolymers. Their rheological behavior was found to be controlled by an interplay of nano-phase separation of the LC units (growth and splitting of their aggregates) and of the thermotropic transitions in these aggregates (which change their stiffness). As a consequence, multiple gel points (up to three) are observed in temperature scans of the copolymers. The physical crosslinks also can be reversibly disconnected by large mechanical strain in the ‘warm’ rubbery state, as well as in melt (thixotropy). The kinetics of crosslink formation was found to be fast if induced by temperature and extremely fast in case of internal self-healing after strain damage. Thixotropic loop tests hence display only very small hysteresis in the LC-melt-state, although the melts show very distinct shear thinning. Our study evaluates structure-property relationships in three homologous systems with elastic PDMS segments of different length (8.6, 16.3 and 64.4 repeat units). The studied copolymers might be of interest as passive smart materials, especially as temperature-controlled elastic/viscoelastic mechanical coupling.

## 1. Introduction

This work is dedicated to ‘smart’ viscoelastic and rheological properties of low-temperature-reversible elastomers, which melt close below room temperature. The materials are based on linear polydimethylsiloxane (PDMS) of different chain length, terminated with liquid-crystalline (LC) units in α- and ω-position. The LC end-groups act as physical crosslinkers in this special variety of PDMS /LC copolymers. To the authors best knowledge, no explicit studies were carried out about materials, which freeze to elastomers at low temperatures (while they are molten at the ambient one), at least not under such a title. On the other hand, numerous studies appeared about low-temperature elasticity, which are dedicated to the behavior of ‘normal’ and most often of commercially important elastomers at low temperatures, see for example, [1]. Fusible rubbers with low melting points offer interesting potential applications (e.g., in soft robotics) as temperature-sensitive smart coupling or energy absorbing materials, which display temperature-switchable viscoelasticity in the melt state, down to ‘freezing’ to a rubbery phase.

In a preceding work (“Part I”, [2]), the synthesis, as well as the phase behavior of the presently studied α,ω-LC-terminated copolymers is subjected to a comprehensive study. In that mentioned publication, the Introduction deals in detail with the specific architecture of the copolymers and especially with the marked difference between them and the ‘liquid-crystalline elastomers’ much-studied in literature: the latter contain a much higher mesogen fraction and thus display wholly different properties.

The study of reversible non-covalent crosslinking via crystallization of small rigid structural units was inspired by the authors’ previous work about epoxy-nanocomposites tethered with polyhedral oligomeric silsesquioxanes (POSS) [1,2,3,4] and about oriented epoxy-LC polymers [5]. In both these systems, however, due to the parallel presence of physical and of relatively dense covalent crosslinking, the possibilities of deeper study of viscoelasticity and rheology were very limited.

### 1.1. Physically Crosslinked Polydimethylsiloxane

Related to the presented study is the broader topic of physical (non-covalent) crosslinking of PDMS. Several interesting approaches were studied in literature, including the attachment of hydrogen-bridging groups [3,6,7,8,9,10] or of π-stacking units [11,12,13,14]. Other physically crosslinked PDMS derivatives include such with grafted side chains consisting of long hydrocarbons with polar end-fragments [15] or A-B-A triblock copolymers with PDMS as central block [16,17]. Combinations of Lewis-acidic and -basic functional substituents on PDMS [18] or of attached mildly bonding ligands and free metal cations [19] also were tested, as well as coordinative crosslinking by borate units [20,21] but such systems rather represent reversible covalent crosslinking. In contrast to the above mentioned so-called ‘thermoplastic elastomers’, the PDMS copolymers studied in the presented work are crosslinked by a rather mild variety of non-covalent interactions: The aggregation of the ‘active’ structural units is driven by their crystallization tendency, with no hydrogen bridging, no strong π-stacking and no strong electrostatic attraction. The authors of this work recently studied [22] a distantly related PDMS–LC copolymer, which, however, is structurally highly different from the presently investigated ones: it consisted of a linear PDMS chain tethered with spatially separated quartets of pendant LC groups. The highly different architecture led to a different thermo-mechanical behavior (compare References [22] and [2]), as well as rheological one (as will be shown in this work).

### 1.2. Rheology of Liquid Crystals and of Liquid-Crystalline Polymers (LCPs)

The presently studied copolymers display highly interesting viscoelastic properties due to the presence of significant amounts of liquid-crystalline building blocks in their structure. The rheology and viscoelasticity of rod-like molecular liquid crystals, as well as of liquid-crystalline polymers (LCPs), attracted a considerable research interest since early on, see reviews [4,5,7,23,24,25,26,27], in view of unusual and promising material properties of LCPs, especially of oriented ones and in view of their extraordinary flow behavior during processing. Lyotropic and thermotropic LCPs generally display similar rheology but the lyotropic ones were studied earlier and more frequently (see e.g., [8,28,29,30]) than thermotropic ones [25], due to the easier in-situ observation of the former by polarized light microscopy (PLM).

#### 1.2.1. Models of Flow Behavior of LCPs

Based on their studies of LCP rheology with polarized light, Onogi, Asada and co-workers [31,32] proposed three regions of the dependence of viscosity on shear rate: a polydomain region with piled and randomly oriented and tumbling LC domains which occurs at low shear rates (“region I”)—associated with thixotropic behavior; at higher shear rates, the “region II” occurs, characterized by dispersed domains in a nematic continuum and by a plateau of the viscosity dependence on shear rate; finally, at the highest shear rates (“region III”), the morphology transforms to nematic monodomain-type (flow alignment) and thixotropic behavior again is observed. Generally, in the aligned nematic state, the LCPs are observed to possess a distinctly lower viscosity than in the isotropic state [23].

The flow behavior of single LCP macromolecules (and also of LC molecules) was described by two models, which apply depending on material properties: The ‘molecular (Doi-) model’ [33,34] which assumes an uniform phase of the flowing fluid with some director wagging or tumbling in this phase. The viscoelasticity arises from the interplay of LC-distortional- (Frank-) elasticity and of hydrodynamic forces. On the other hand, the ‘polydomain (Larson-Doi-) model’ [35,36] is based on a polydomain morphology (similar to the polycrystalline one in solid materials), where not the director in the uniform fluid but the small domains undergo wagging and tumbling [37]. If the rheological behavior of lyotropic and thermotropic LCPs was compared [38,39], it was found, that the molecular (Doi) theory well fits diluted lyotropic LCPs, while concentrated lyotropic LCPs and thermotropic LCPs are better described by the polydomain model (Larson-Doi). The latter is favored by topological exclusion and insufficient flexibility of the macromolecules.

#### 1.2.2. Effect of LCP Architecture

Several studies compared the rheology of the two main structural families of LCPs:

In case that thermotropic main-chain-LCPs (MC-LCPs) possess semi-flexible properties due to suitable spacers, flow aligning behavior was found to be greatly favored [27,40], even in a moderately crosslinked MC-LCP [41]. Because of this, MC-LCPs are also subjected to rather durable consequences of previous shear flow (‘thermal and shear history’), which are difficult to reset, even by isotropic melting [27,42,43].

The side-chain-LCPs (SC-LCPs) were reported to possess a much more extended linear viscoelastic region than MC-LCPs [44] and even to lack the rheological ‘region I’ [42], as well as having a tendency to persistent domain tumbling during flow [27], instead of flow alignment. In contrast to MC-LCPs, shear history effects quickly disappear in SC-LCPs (rapid ‘reset’ of initial viscoelastic properties) [27]. SC-LCPs with the rod-like mesogens attached side-on (laterally) [45] are generally similar to ‘normal’ SC-LCPs with end-on-attached LC units but they can be oriented by creep, while subsequent isotropization easily occurs via oscillatory deformation. Their rheology is controlled mainly by their backbone.

Spacer moieties in MC-LCPs influence their tendency to flow alignment by their flexibility [27,40], as well as their viscosity (odd/even effect in shorter polymethylene spacers) [46]. In SC-LCPs, a longer spacer favors stronger LC-LC interaction and eventually the formation of a smectic phase [47,48], while shorter spacers favor nematic ordering. Smectic SC-LCPs display a specific shear behavior [49].

### 1.3. Rheology-Related Studies on α,ω-LC-Terminated Polymers

The α,ω-LC-terminated PDMS polymers studied in this work belong to a unique structural group, distinct from SC-LCPs and MC-LCPs. In case of a not too long central chain, such compounds were called ‘LC-dimers’ in older literature. Early examples of rheological investigations on such compounds (which however contained different central chains than PDMS) are the publications [50] and [51,52]. A relatively small degree of segmental phase separation (LC/central chain) was achieved in Reference [50] and in most products in Reference [51], which reduced the ‘smart’ properties of such ‘LC-dimers’. In Reference [51], smectic ordering was achieved in one of the ‘LC-dimers’, via suitable choice of the central chain. In Reference [52], polytetrahydrofuran-28-mer α,ω-terminated with an aromatic diester (“diES2-polyTHF”), was subjected to comprehensive rheological investigation. In contrast to molecular liquid crystals or to MC-LCPs, diES2-polyTHF displayed a higher viscosity in the nematic state than in the isotropic melt, obviously due to supramolecular assembly (nano-phase-separation of LC). Large strains were found to induce solid → liquid transition in diES2-polyTHF (in contrast to MC-LCPs). After shear treatment or large oscillatory deformation, the recovery of original material properties was very slow, which was attributed to a slow build-up of nano-phase-separated domains. Flexibility and nano-phase-separation tendency of the central chain were considered to be the key factors in the smart behavior of “diES2-polyTHF” in Reference [52]. In the present work, these both parameters of an α,ω-LC-terminated copolymer were much improved, which led to very interesting rheological behavior.

#### 1.3.1. Rheological Properties of Physically Crosslinked PDMS

One of the first viscoelastic studies on PDMS-LC copolymers (of loosely crosslinked SC-LCP) was done by Finkelmann and co-workers [53]. Some persistent shear-induced anisotropy was observed, as well as ‘liquid-crystalline elasticity’ upon isotropic → nematic transition. More related to the studied copolymers are the PDMS polymers α,ω-terminated with various physically crosslinking groups, which were mentioned further above. As these products displayed intriguing properties in many aspects, their rheology or viscoelasticity was studied only to a moderate extent, in the further-above cited works [7,8,13,18]. Similar was also the situation with rheological characterization of the less structurally related PDMS physically crosslinked by pendant groups [12,14,54,55]. All the mentioned works contain rather simple rheological characterizations like the study of the melting process by recording temperature-dependent moduli [7] or viscosity [13]. More sophisticated investigations included stress relaxation tests which characterized the re-organization of the reversible physical crosslinks and its dynamics [14,54,55], as well as deformation recovery tests [54].

#### 1.3.2. Aim of This Work

The aim of the presented work was a comprehensive study of the complex viscoelastic and rheological properties of the α,ω-LC-terminated PDMS copolymers recently prepared by the authors, in the molten, as well as in the rubbery state and also the evaluation of structure property relationships in this context. While the rheology and viscoelasticity of typical liquid crystalline polymers was studied to a considerable depth, the presently investigated materials are novel, due to their low volume fraction of LC units. Their rheology and viscoelasticity is expected to be highly different, dominated by the interplay of nano-phase-separation effects, of the elasticity of the PDMS chains and by the thermotropic behavior of the physical crosslinks, made up of LC aggregates. The studied low-temperature-melting elastomers are interesting because of potential applications as thermo-responsive viscoelastic coupling materials, behaving as oils with greatly varying temperature-controlled viscoelasticity near room temperature and as rubbers at lower temperatures. An attractive feature of the studied products are their azo LC groups, which offer development potential for photo-sensitivity.

## 2. Experimental Section

### 2.1. Materials

In this work, the rheological properties of three α,ω-mesogen-terminated copolymers were studied, which consisted of polydimethylsiloxane (PDMS) central chains and of diaromatic azo mesogen end groups (called “BAFKU” in continuation of nomenclature from previous work). The copolymers differed in the length of the central PDMS chain and were named “H03–BAFKU_2_”, “H11–BAFKU_2_” and “H21–BAFKU_2_” (where “Hxx” denoted the different central chains). The synthesis and basic characterization of these materials is described in a preceding work by the authors (“Part I” of the present one) [2]. As reference compounds for some of the tests, the neat PDMS spacer components were also employed, namely “DMS H03” (*M*_n_ = 623.9 g/mol), “DMS H11” (*M*_n_ = 1196.5 g/mol) and “DMS H21” (*M*_n_ = 4764 g/mol), which were purchased as commercial products (Gelest, Inc., Morrisville, PA, USA).

### 2.2. Rheological Characterization of the PDMS–BAFKU_2_ Copolymers

#### 2.2.1. Equipment

The advanced multi-functional rheometer of the type ARES-G2, from TA Instruments, New Castle, DE, USA—part of Waters, Milford, MA, USA, was used for characterizing the rheological properties of the prepared copolymers. Different experimental procedures were carried out on this rheometer, in order to perform the below-described investigations.

#### 2.2.2. Sample Geometry

The samples were measured between parallel plates, the diameter of which was: 35 mm for BAFKU2-H21, 25 mm for BAFKU2-H11 and most experiments with BAFKU2-H03 and 12.6 mm for the most viscous samples of BAFKU2-H03. Pure polysiloxane components H03, H11, H21 were measured as references using the 12.6 mm plates. The thickness of the tested samples was always between 0.2 and 0.25 mm.

#### 2.2.3. Determination of Gel Points in Multi-Frequency Oscillatory Tests

The temperature-controlled gelation behavior of the studied copolymers was evaluated using the multi-frequency temperature sweep test (method name in ARES-G2 software: “Oscillation Multiwave”): the samples were loaded as melt between parallel plates, the geometry of which is specified above and were izotropized by a 5 min dwell at 80 °C. Subsequently, two tests were run, the first one in cooling regime and a second one in heating regime. The first test started at the ‘loading temperature’ of +80 °C, where all samples were isotropic and all history effects in them erased. The final temperature of the cooling run depended on the sample: 0 °C for H03–BAFKU_2_, −15 °C for H11–BAFKU_2_ and −40 °C for H21–BAFKU_2_. The heating scan started at the final cooling temperature of each sample and ended at +70 °C. The rate of temperature change was 1 °C/min in both scans. The deformation regime in both scans consisted in simultaneously applied (‘multi-frequency’) deformations of following frequencies and amplitudes: 1 Hz/strain amplitude of 1%, 2 Hz/1%, 4 Hz/0.8%, 8 Hz/0.6%, 16 Hz/0.4%, 32 Hz/0.3% and 64 Hz/0.1%. The results were depicted as curves sets of temperature dependent storage shear modulus *G*′ = f(*T*), of the loss shear modulus *G*″ = f(*T*) and of the loss factor tan(δ) = f(*T*). The latter sets of curves were used to determine the gel points (as tan δ crossover points) according to the theory of Chambon and Winter [56].

#### 2.2.4. Rate of Thermally Induced Physical Gelation

In order to evaluate the rate with which the physical network is formed after abrupt cooling of molten copolymer to different temperatures, oscillatory time sweep tests (method name in ARES-G2 software: “Oscillation/Time”) with a constant oscillatory frequency of 1 Hz and a constant strain amplitude of 1% have been performed. The samples were measured between parallel plates, the geometry of which is specified further above. The tested copolymer samples were rapidly cooled from +70 °C in the isotropic molten state to different final temperatures, which were positioned in the rubbery region, in the liquid-crystalline melt region and in the isotropic melt region. The “oscillatory time sweep test” recorded *G*′, *G*″ and tan(δ) as function of time, as well as the actual temperature, beginning with the start of the rapid melt cooling. In this way, eventual rapid crosslinking prior to temperature equilibration also could be observed, in addition to slower crosslinking which occurred after the temperature of the cooled melt stabilized.

#### 2.2.5. Analysis of Mechanical Disconnection of Crosslinks: Strain Sweep Tests

The possibility to disconnect the reversible physical networks by mechanical strain was investigated using the oscillatory strain sweep test (method name in ARES-G2 software: “Oscillation Amplitude”). The samples were measured between parallel plates, the geometry of which is specified further above. At a constant deformation frequency of 1 Hz, the strain was gradually increased from 0.1% to 1000% and strain-dependent *G*′, *G*″ as well as tan(δ) were recorded. Such experiments were carried out at several constant temperatures positioned in characteristic regions of the studied samples.

#### 2.2.6. Frequency Stiffening Tests

Eventual sample stiffening at high oscillatory deformation frequencies was studied by means of the frequency sweep test (method name in ARES-G2 software: “Oscillation Frequency”), where the oscillatory deformation frequency was gradually increased from 0.001 Hz to 100 Hz. The data were recorded in five logarithmic series of points—‘decades’, with a constant deformation amplitude in each decade, namely 50% (at 0.001 to 0.01 Hz), 20%, 10%, 5% and 1% (at 10 to 100 Hz). The samples were measured between parallel plates, the geometry of which is specified further above. Such frequency sweep tests were carried out at several constant temperatures positioned in characteristic regions of the studied samples.

#### 2.2.7. Creep and Creep Recovery Tests

The creep and creep recovery tests were performed in order to further evaluate the strength of the physical crosslinks in the studied materials at different temperatures. In weaker-crosslinked samples, yield stress values could be obtained in these experiments. The samples were measured in the torsion regime between parallel plates, the geometry of which is specified further above. The method “Step (Transient) Creep” (method name in ARES-G2 software) was used for this purpose. The experiment consisted of several subsequent stages, each of which contained a loading step followed by an un-loading one (two subsequent “Step (Transient) Creep” modules). During the loading step, a constant stress was applied for a pre-defined time period and the deformation (strain) adjusted automatically in order to maintain the constant stress. In the subsequent un-loading step, the value of the applied constant stress was set to be zero. The measured time-dependent values of the automatically adjusted strain were the final results. The subsequent two-step stages differed in the strain values applied in their respective first step. The creep and creep recovery tests were performed at several characteristic temperatures for each studied copolymer. The standard sequence of applied stress loadings (in the first steps of each stage) was: 0.1, 1, 200, 500, 1000, 2000 and finally 10,000 Pa. In view of the properties of the studied samples, the duration of the loading and un-loading steps was set to 3 min, with 200 data points recorded during this time.

#### 2.2.8. Stress Relaxation Tests

Stress relaxation tests (method name in ARES-G2 software: “Step (Transient) Stress Relaxation”) were used as an additional method to further evaluate the strength of the physical crosslinks and the dynamics of their splitting and reconnection. The samples were measured in the torsion regime between parallel plates, the geometry of which is specified further above. During each test, a pre-defined constant deformation (strain) was applied and the time-dependence of the resulting stress was recorded by the force detector and later evaluated as the final experiment result. The experiment was stopped after the stress value apparently reached equilibration. For each copolymer sample, a series of tests was done at several characteristic temperatures. Each of the series consisted of tests with the applied deformation (strain) values of: 0.4%, 1%, 2%, 3%, 4%, 5%, 10% and 20%. The data sampling rate was 1 point/s.

#### 2.2.9. Simple Oscillatory Tests of Thixotropy

The destruction of physical crosslinking by high deformations, as well as its recovery upon substantial reduction of such deformations was tested in simple oscillatory time sweep tests (method name in ARES-G2 software: “Oscillation/Time”). The experiments were carried out as multi step procedures. In each step, a constant oscillatory frequency (1 Hz) and at constant oscillatory deformation was applied and the time-dependent values of storage shear modulus and of loss shear modulus were recorded. The individual steps in the procedure differed by highly contrasting values of the deformation amplitudes, which ranged between 0.1 and 5030%. The duration of each step was between 50 and 150 s.

#### 2.2.10. Thixotropic Loop Tests

Thixotropy loop tests were performed in order to evaluate changes in viscosity caused by continuous shear flow, as well as the degree of eventual recovery vs. ‘shear damage’ in short-time term. The samples were measured between parallel plates, the geometry of which is specified further above.

Two experimental setups were used: “Flow Ramp” and “Flow Sweep” (method names in ARES-G2 software):

In the “Flow Ramp” tests, in the first step, the materials were subjected to a continuously increasing shear rate (during continuous rotatory mode) while the stress generated by the shearing was recorded. The shear-rate-dependent viscosity values were calculated from form the stress values. In the second step, the shear rate was continuously reduced down to 0 s^−1^. In a standard test, the range of applied shear rates was 0 to 100 s^−1^, followed by 100 to 0 s^−1^. The duration of each step was set to 10 min. In case of highly viscous or semi-solid samples, shorter “Thixotropy ramp” tests were performed, with shear rate ranges of 0 to 0.1 s^−1^, 0 to 1 s^−1^ or 0 to 10 s^−1^. The values of stress and of viscosity in dependence of shear rate were obtained in the result plot.

The “Flow Sweep” tests were used to perform a rapid scan of the shear rate region from 100 to 400 s^−1^. The experiments were performed in two steps: In the first step, the shear rate was step-wise increased, with measured points at: 100, 200, 300 and 400 s^−1^ and with equilibration time of 2 min per each point (while the point’s data were measured and averaged for an additional 30 s). In the second analogous step, the shear rate was step-wise decreased, to 300, 200 and 100 s^−1^. The values of stress and of viscosity in dependence of shear rate were obtained in the result plot.

The thixotropy loop tests were carried out at several characteristic temperatures of each copolymer, in order to characterize their semi-liquid and liquid regions.

## 3. Results and Discussion

### 3.1. Description of the Studied Copolymers

Three copolymers (structure: Scheme 1) consisting of highly flexible polydimethylsiloxane (PDMS) chains, α,ω-terminated with mesogenic units of azobenzene type (“BAFKU”), were subjected to a comprehensive rheological characterization as potential passive smart materials. The copolymers differed in the length of the central PDMS chain, namely 8.6-mer (“H03”), 16.3-mer (“H11”) and 64.4-mer (“H21”). Their synthesis and basic characterization was described in a previous work [2] by the authors. From the point of view of nomenclature, however, the most proper designation of the studied products would be “mesogen end-capped polymers”.

The combination of the highly flexible central chain with two rigid and PDMS-incompatible mesogenic BAFKU groups leads to nano-phase-separation and to physical crosslinking (see Scheme 2) via the aggregation of BAFKU end-groups (nano-crystallization). The latter effect was found to result in the formation of a distinct lamellar morphology in the copolymers [2], which also persists in the lower-temperature region of the copolymer melt, until the regular lamellae break-up above the isotropization temperature [2] (see Scheme 2). Thermotropic transitions of the liquid-crystalline (LC) end-groups assembled in the lamellae were expected to cause interesting viscoelastic and rheological behavior of the studied materials, that is, melting vs. solidification of the rubbery phase or temperature-induced changes in viscoelasticity of the melt.

In Scheme 3, Scheme 4 and Scheme 5, the previously determined [2] thermal characteristics of the studied copolymers are summarized, which will be important for the evaluation of their rheology:

The H03–BAFKU_2_ copolymer displays the richest phase behavior [2] (Scheme 3): In the solid state it is glassy, then, upon heating, this vitrimeric material transforms directly from glass to melt. In the liquid state, first a smectic (Sm) phase forms, followed by a nematic (N) one and finally by the isotropic (I) melt phase, in the order of increasing temperature. The lamellar structure, as shown in Scheme 2, exists in the solid (glassy) state, where the ordering of the BAFKU units in the lamellae is crystalline (Cr), as well as in the Sm and N state of the melt. The Cr → Sm and Sm → N transitions occur inside of the lamellae, while the N → I transition is connected with lamellae breakup to irregular and dynamically splitting/recombining ‘nano-droplets’ (final superstructure in Scheme 2). The H03–BAFKU_2_ copolymer also is characterized by the highest volume fraction of the LC units (“BAFKU”), namely 52%.

The H11–BAFKU_2_ copolymer somewhat differs from H03–BAFKU_2_ [2] (Scheme 4) but it also is characterized by the same lamellar structure (see Scheme 2). This copolymer is glassy at the lowest temperatures, thereafter it transforms into a rubbery phase, which subsequently melts to a nematic liquid (without undergoing a smectic state) and it finally transforms into an isotropic liquid, in the order of increasing temperature.

H21–BAFKU_2_ (Scheme 5) generally was found [2] to be very similar to H11–BAFKU_2_, except for the values of the characteristic temperatures. The DSC transitions were much less intense in the longest copolymer: they were partly below DSC detection threshold.

The longer copolymers are dark red viscoelastic liquids at standard room temperature (25 °C), with melting/freezing points at +10/+5 °C (H11–BAFKU_2_) and −12/−13 °C (H21–BAFKU_2_). The mesogen-rich H03–BAFKU_2_ is turbid (smectic) sticky and highly viscoelastic paste of orange color at 25 °C (dark red in nematic and isotropic melt) and its melting/freezing points are at +23/+11 °C.

The viscoelastic and rheological properties of the studied non-covalently crosslinked copolymers in the rubbery and in the molten state were investigated by multiple methods, while focusing on four aspects: First, the gelation behavior was studied at the transition melt/solid, as well as during thermotropic transitions in the melt. Secondly, the kinetics of thermally induced gelation to solid (or to partly crosslinked viscoelastic liquid) was characterized. Thirdly, the mechanical reversibility of the non-covalent crosslinks also was evaluated. Finally, the shear thinning (thixotropic) behavior was investigated, which also is related to the mechanical disconnection of crosslinks.

### 3.2. Reversible Gelation Processes Induced by Temperature Change

The thermally-induced gelation of the studied copolymers was investigated in temperature areas stretching from temperatures somewhere below their respective solid/melt transition, up to the region of isotropic melt. Temperature scans over the whole interesting range were carried out, in multi-frequency oscillatory deformation mode. The experiments were performed both in the cooling- as well as in the heating regime, in order to compare the inverse processes of crosslink formation and dissociation. As will be demonstrated further below, both the effects of thermotropic transitions in LC-nano-aggregates, as well as the wider-scale effects of nano-phase-separation (LC-lamellae fragmentation vs. growth) appear to play an important role in the gelation behavior. Generally, multiple gel-points were observed in the copolymers, up to three in case of H03–BAFKU_2_. As a point of gelation, the point (the temperature) of *tan δ* crossover was sought (point of frequency-independence of *tan δ*), in accordance with the theory of Chambon and Winter [56].

H03–BAFKU_2_: The results of the ‘multi-frequency gelation tests’ obtained for the H03–BAFKU_2_ copolymer are shown in Figure 1: In Figure 1a,b, the temperature-dependence of the storage shear modulus (*G*′: blue curves), of the loss shear modulus (*G*″: green curves), as well as of the complex viscosity (*Eta**: black curves) is shown. Curves of each temperature-dependent magnitude were recorded at different frequencies (1, 2, 4, 8 and 16 Hz) and the order of the curves for these frequencies is denoted in the graphs. Figure 1a summarizes the results from the cooling run and Figure 1b the heating run (both at 1 °C/min). These graphs illustrate the temperature- and the frequency-dependence of the basic viscoelastic properties. Figure 1c,d and display the temperature-dependence of the loss factor (*tan δ*) alone, which was equally measured at the same multiple frequencies. Figure 1c shows the cooling run and Figure 1d the heating run. These latter graphs were used to find the temperature-induced gel points of the studied material.

If the basic viscoelastic characteristics of H03–BAFKU_2_ in the melting and in the low-temperature-melt region are evaluated (Figure 1a,b), it can be noted that this copolymer displays a 40 °C-wide region of thermally-induced transitions, where storage and loss moduli, as well as the viscosity change by many orders of magnitude, in several ‘gradual steps’. At lower- (vitrimeric solid), as well as at higher temperatures (isotropic melt), the region of dramatic changes is bordered by ‘plateau regions’, where the mentioned magnitudes undergo little to moderate temperature-induced changes. The results recorded in the heating and in the cooling regime somewhat differ: A visible temperature shift is observed (delayed build-up of moduli and viscosity in the cooling run), which only partly correlates with the rather small shift (heating/cooling) in the temperatures of thermotropic transitions of H03–BAFKU_2_ which were determined by DSC in Reference [2] and which are included as markers in Figure 1. Also, the curves’ shapes are somewhat different. Different dynamics of growth vs. fragmentation of nano-phase-separated lamellae might strongly contribute to the observed undercooling effect. It can be noted, that the complex viscosity is strongly frequency-dependent in the vitreous state and visibly frequency-dependent in the liquid-crystalline melt, while the frequency-dependence practically disappears in the isotropic melt.

The multi-frequency sets of *tan δ* = f(*T*) curves of H03–BAFKU_2_ (see Figure 1c,d) indicate an interesting gelation behavior, with up to three gel points, both in the cooling and heating scan. Close to the solid/liquid transition, where a gelation would be expected, a narrowing (with *tan δ* curves nearly ‘touching’ in the cooling run) or a true crossover (in the heating run) of the curves indeed is observed. This gel point precedes a distinct *tan δ* maximum, which occurs at a somewhat higher temperature. The mentioned two features appear to be directly connected with the Cr/Sm transition in the lamellar aggregates of BAFKU end-groups: The gel-point marks the solid/liquid transition, while the maxima in *tan δ* can be assigned to energy absorption by friction of smectic layers in the lamellar aggregates (see Scheme 6 and Scheme 7).

At even higher temperatures, a distinct *tan δ* crossover is observed in both runs. This second gel point is can be correlated with the Sm/N transition (reported in Reference [2] and marked in Figure 2), which occurs at a temperature slightly below the second gel point. The physical reason for this gelation likely is (in the cooling run) the increased spatial crosslinking, via the transition from 1-dimensionally ordered nematic lamellae to 2-dimensionally ordered smectic lamellae (see Scheme 6) and possibly also the growth of more rugged lamellar superstructure upon this transition. (An opposite process occurs in the heating run).

Finally, most likely a third gel point occurs at the temperature approaching the one of the N → I transition (reported in Reference [2] and marked in Figure 2), although the sensitivity of the experiments was not high enough to confirm definitely this third crossover. This gel point would correspond (upon cooling) to the formation of 1-dimensional crosslinking in isotropic droplet-like BAFKU nano-domains (see Scheme 6) and probably also to the growth of much more robust nematic-ordered lamellar nano-aggregates, starting from droplet-like liquid ones. (An opposite process occurs in the heating run).

H11–BAFKU_2_: The ‘multi-frequency gelation results’ obtained for the longer H11–BAFKU_2_ copolymer are shown in Figure 2 (changes in basic viscoelastic properties in Figure 2a,b; analysis of gel points—*tan δ* crossovers in Figure 2c,d), recorded both as cooling run (Figure 2a,c) and as heating run (Figure 2b,d). The H11–BAFKU_2_ copolymer displays similar trends like the shorter H03–BAFKU_2_ but a notable difference is the absence of a smectic phase: Upon melting of the whole material, as well as of the BAFKU lamellae, the ordering in the latter transforms from crystalline to nematic (as reported in Reference [2]). A step is observed in the basic viscoelastic magnitudes at this temperature (Figure 2a,b), as well as a *tan δ* crossover (Figure 2c,d). This transition, similar to the corresponding DSC peak (reported in Reference [2] and marked in Figure 2) undergoes a strong undercooling in the cooling scan. The gelation to rubber/rubber melting (also Cr/N transition) is followed by a large maximum in *tan δ*, which can be assigned to energy absorption in the lamellar aggregates of BAFKU: In the heating scan, the mesogenic units become more mobile and additionally the lamellae more easily undergo splitting (see Scheme 8).

A second gel point (*tan δ* crossover) as well as a distinct step in the storage modulus (and a flat step in viscosity and loss modulus) is observed near 45 °C, at a temperature close to the DSC peak of the N/I transition (reported in Reference [2] and marked in Figure 2). A similar N/I gel point was also observed in the copolymer H03–BAFKU_2_. Similarly like in the latter case, there is only a small undercooling effect, if comparing the cooling and heating run of this transition. The *tan δ* crossover is well-visible in the cooling scan but its precise position might be influenced by thixotropy effects resulting from changed strain amplitudes (highlighted by red markers on the Temperature axis in Figure 2; the amplitude was changed in order to compensate for strongly decreasing sample resistance, in order to keep signal quality). Such thixotropy effects are distinctly visible on several occasions in Figure 2. A distinct difference between the heating and cooling scans are the markedly lower values of *tan δ* in case of the cooling scan. This might be attributed to the mentioned thixotropy effects, which were different in case of cooling- and heating scan, because the cooling scans started with larger deformations.

H21–BAFKU_2_: The longest copolymer generally displays nearly analogous features in the multi-frequency gelation tests, like the other ‘long copolymer’, the above-discussed H11–BAFKU_2_, albeit with different characteristic temperatures. The results are shown in the Appendix A. H21–BAFKU_2_ contains only 13.5 Vol.% of the mesogen units and hence the physical crosslinking via BAFKU aggregation in the nematic melt region (or more precisely region of where the BAFKU aggregates are nematic) is less efficient and the moduli *G*′ and *G*″ are markedly lower than in case of H11–BAFKU_2_. The *tan δ* curves in the melt region are thus very noisy but nevertheless, a *tan δ* crossover in the region of the melting of the rubbery phase can be discerned, especially well in case of the heating run (Appendix A). An adjacent region with *tan δ* maxima (similar like in H11–BAFKU_2_) still can be recognized, albeit with some difficulty (heating run, Appendix A). The suspected second *tan δ* crossover in the region of isotropization near 30 °C (invisible by DSC but clearly indicated by PLM in Reference [2]) cannot be discerned but the associated step in the *G*′ = f(*T*) curves still can be seen, if the deformation amplitude is carefully selected (Appendix A). Thixotropy effects are very strong in the melt of H21–BAFKU_2_, which is attributed to a higher lability of the BAFKU aggregates, which are more dispersed in this copolymer. These effects can be seen especially well if comparing the sets of *tan δ* curves obtained for two different cooling runs carried out with strongly different deformation amplitudes, as shown in Appendix A vs. Appendix A.

### 3.3. Rate of Physical Gelation upon Quenching the Melt

An interesting aspect of the temperature-induced reversible crosslinking of via nano-aggregation of BAFKU units in the studied copolymers is the kinetics of this process. This could not be directly observed in the above-discussed experiments.

The kinetics was studied via ‘quenching’ polymer melt samples from the ‘isotropic temperature’ of 70 °C down to temperatures ranging between −50 and +60 °C. During the quenching and in the subsequent time period where the temperature was constant, the moduli of the tested samples were measured via an applied oscillatory deformation of constant frequency (1 Hz) and of a constant small strain (1%). Observation of the build-up and of the subsequent equilibration of the moduli gave a picture of the kinetics. The complete sets of results for all the three copolymers are shown in the Appendix A.

In general it was observed, that the temperature-induced gelation was relatively fast, with equilibration times ranging between 0.5 and 5 min. In case of slower gelations, especially of H21–BAFKU_2_, it was possible to recognize that the gelation was a two-step process, starting with the rapid formation of small ordered aggregates of BAFKU end-groups from droplet-like isotropic liquid ones and followed by the slower growth of the aggregates to larger ones. Hence, the crosslinking process is not only connected with thermotropic (LC) transitions but also with the dynamics of nano-phase-separation. All the copolymers displayed similar trends.

H11–BAFKU_2_: Selected results concerning the kinetics of the gelation of the isotropic melt of this copolymer upon quenching to different lower temperatures are shown in Figure 3. (All results are shown in Appendix A). The graphs in Figure 3 contain the time-dependent course of the storage and loss moduli (*G*′ and *G*″ respectively), as well as the course of the temperature of the sample. The equilibration time is short at the lowest quenching temperatures (40 s at −50 °C) but it gradually increases (see Appendix A). At +20 °C, it already is 240 s (6 min). The time value contains also the time needed for temperature equilibration, ca. 40 s in case of the +20 °C experiment in Figure 3. For final temperatures above +20°C, a distinct induction period appears, during which the moduli do not change and which makes most of the difference between the gelation times at the different final temperatures higher than +20 °C. This induction time is significantly longer than the time needed for temperature equilibration (see Appendix A). The increasing induction time likely indicates the more difficult onset of LC lamellae growth (see Scheme 9) at higher final temperatures.

If the shape of the curves *G*′ = f(*time*) and *G*″ = f(*time*) is compared, similar courses are obtained if the ‘hot melt’ is frozen to a (rubbery) solid or only to a viscoelastic melt. In the latter case, the mentioned induction time prior to melt thickening appears as an additional feature. Also, the curves for *G*′ have very similar courses and are always close to the ones of *G*″ but in case of rubber as final state, *G*′ has a higher value than *G*″, while the opposite case is found if cooling down to viscoelastic melt only.

The H21–BAFKU_2_ copolymer displays fairly analogous gelation kinetics like H11–BAFKU_2_ (see selected results in Figure 4 and all results in Appendix A) but the gelation process of H21–BAFKU_2_ clearly displays a two-step character (see Figure 4, well visible above 4−0 °C). This can be assigned to a rapid formation of small ‘primary’ nano-aggregates of the BAFKU end-groups, which subsequently more slowly (second step in the curves) grow to larger, higher-functional-crosslinking ones (see Scheme 9). Induction periods prior to crosslinking are the longest in H21–BAFKU_2_ among the tested copolymers, as well as the times of moduli equilibration. This appears to be connected with the very long PDMS spacer segments and with the hence more difficult assembly of the domains which act as crosslinkers.

H03–BAFKU_2_, a vitrimer, was tested only in its viscoelastic melt region. The gelation kinetics results are shown in Appendix A. It shows analogous trends to the ones observed for H11–BAFKU_2_. A difference is, that in the smectic liquid region near the melting point (30 °C) of H03–BAFKU_2_, the physical crosslinking achieves a much higher relative modulus increase (3 orders) than the nematic crosslinking in case of H11–BAFKU_2_, although the absolute values of equilibrium moduli of both copolymers in the melting region are similar. At 30 °C, the sample H03–BAFKU_2_ additionally displays visible thixotropy during the kinetics test: moduli somewhat decrease after initial build-up, as consequence of the oscillatory shear deformation of 1% amplitude. Another specific feature of H03–BAFKU_2_ are the much shorter or even absent induction periods, if the isotropic melt is cooled down to temperatures higher than 30 °C. The longer times of equilibration are due exclusively to crosslinking. The absence of induction times in the ‘viscoelastic gelation’ of H03–BAFKU_2_ can be attributed to the dominant (over 50%) volume fraction of the BAFKU mesogen, which facilitates the start of lamellae assembly.

### 3.4. Mechanical Disconnection of the Physical Network

After studying the thermal reversibility of the physical crosslinking via BAFKU units nano-crystallization, their resistance against mechanical disconnection was investigated. Several experimental methods were employed to this end: Short-time stability at small and high deformations was studied in strain sweep tests carried out at different characteristic temperatures. Additionally, creep tests and stress-relaxation tests were performed, in order to evaluate the crosslink stability on a moderately long time scale. The results indicated a high strength and durability of the non-covalent crosslinks in the rubbery state, if the temperature was sufficiently far below the melt solidification temperature. While approaching this temperature, the mechanical disconnection of the crosslinks became increasingly prominent.

#### 3.4.1. Strain Sweep Tests

The strain sweep tests were carried out in oscillatory mode, at the frequency of 1 Hz. The applied deformation amplitudes were changed from 0.1% up to 1000%. Representative results, as obtained for H11–BAFKU_2_ are shown in Figure 5.

All the studied copolymers display similar trends, especially close are the two longer ones, the elastomers H11–BAFKU_2_ and H21–BAFKU_2_. The vitrimer H03–BAFKU_2_ could be investigated only in the viscoelastic melt region, where it mostly shows analogous trends like the remaining copolymers. The collection of results for all the copolymers is shown in Appendix A.

In case of H11–BAFKU_2_, which is shown as an exemplary system in Figure 5 (all data are in Appendix A), it can be noted, that at very low temperatures far from the melting region (e.g., −50 °C in Figure 5) the physical network in the rubbery phase is rather robust: The sample undergoes macroscopic destruction at sufficiently high strains, which manifests itself as irregularities and sudden downward steps in the graphs. The mechanical destruction can also be observed visually. This damage occurs practically without dynamic disconnection of the physical crosslinks. At somewhat higher temperatures (−30 °C in Figure 5), a smoother destruction occurs, which besides macroscopic damage likely also includes some shear thinning in the material (crosslink disconnection). At even higher temperatures (−20 °C in Figure 5) no mechanical destruction is observed any more visually, the measured curves are smooth and the samples are able to endure even high deformations. The moduli smoothly decrease at sufficiently high deformations (100% and more at −20 °C in Figure 5), due to the disconnection of the physical crosslinks. At such a high deformation, the sample becomes liquid-like as *G*″ exceeds *G*′ (*G*″ *G*′ crossover). At even higher temperatures, for example, 0 and 10 °C in Figure 5, the strain dependence of the moduli shows similar trends like before but *G*″ always is higher than *G*′, as the material already is a melt and also the ‘shear degradation’ of the moduli starts at lower strains. The curves also become flatter and the moduli smaller.

H21–BAFKU_2_ (see Appendix A) differed from H11–BAFKU_2_, only in the characteristic temperatures (really rigid crosslinks at −70 °C or lower),

The vitrimer H03–BAFKU_2_ (see Figure 6 and all data in Appendix A) displays some interesting differences in the early post-melt state, which is a smectic phase: At 25 °C (see Figure 6), the melt is highly viscoelastic and the storage modulus *G*′ even exceeds *G*″ at the lowest deformation amplitudes (below 1%). After a small plateau until the strain amplitude of ca. 1%, both moduli then display marked shear thinning, *G*′, much more so than *G*″. This shear thinning behavior is similar to the one observed by Yang and Chang [49] for a well-characterized smectic polisiloxane-LC copolymer, albeit the latter had a different architecture than H03–BAFKU_2_ (it was a side-chain LC-copolymer very rich in LC). At higher temperatures, H03–BAFKU_2_ displays similar trends in the viscoelastic melt like the other copolymers.

#### 3.4.2. Creep Tests

The mechanical strength of the physical crosslinks also was studied in multi-step creep experiments, in which a constant stress was applied in each step and the resulting time-dependent strain was observed. After each loading step, a recovery step with stress automatically adjusted to zero was carried out. The copolymers displayed similar trends in their respective characteristic temperature regions. The crosslinks were found to be robust only in the low-temperature part of the rubbery region.

H11–BAFKU_2_: Representative results for the copolymer H11–BAFKU_2_ are shown in Figure 7, while the complete collections of results for all the three copolymers can be seen in Appendix A. It can be seen, that in the low-temperature part of the rubbery region (−50 °C in Appendix A) nearly no creep (in view of the error margin) occurs on the time scale of 3 min at any applied stress (up to 10,000 Pa). In the warmer temperature region of the rubbery phase (−20 °C in Figure 7), distinct creep starts at 10,000 Pa and is no more fully recovered. At −5 °C, not far from the melting region, creep is observed already at the lowest loadings, although significant elastic recovery still occurs, especially at lower stress loadings. By applying 10,000 Pa at −5 °C, the sample is ‘shear-liquefied’ and an extreme deformation results: see inlay with different y-axis scale in the graph on the right side in Figure 7. This extreme deformation is plastic and does not significantly recover. At +5 °C and at higher temperatures, only plastic deformation without any recovery occurs and the strains resulting from the applied stresses become very large, as the sample becomes liquid, albeit a nematic viscoelastic one.

H21–BAFKU_2_: The longest copolymer which displays the lowest melting point of the rubbery phase, H21–BAFKU_2_, generally displays analogous trends like the shorter H11–BAFKU_2_ (see results in Appendix A). In the rubbery region at low applied stresses, the creep tendency of H21–BAFKU_2_ is smaller than in case of H11–BAFKU_2_ but at the highest stresses, it is the longer copolymer which creeps more. This might be the combined effect of more numerous elastically active entanglements in H21–BAFKU_2_ (due to longer chains) on one hand, as well as of smaller and hence less strong BAFKU aggregates (physical crosslinker). In this context, it is interesting that in the early post-melt nematic state, the sample still exhibits a considerable elastic recovery and starts to flow only if 2000 Pa or more are applied.

H03–BAFKU_2_: The shortest copolymer, the vitrimer H03–BAFKU_2_, could only be accurately rheologically investigated in the melt region. The results in Appendix A clearly show, that from +15 °C upwards, the melting material already is fully plastic, with no significant elastic recovery.

#### 3.4.3. Stress Relaxation

The mechanical stability of the physical crosslinks was further studied by means of stress relaxation tests, which practically evaluated ‘internal creep phenomena’: A sample, in which the crosslinks are dynamic, eventually would fully adjust to the new deformed shape via disconnection and more favorable recombination of crosslinks, behaving similarly like a viscoelastic liquid. All the copolymers displayed similar trends in their respective characteristic temperature regions, like in case of creep tests.

H21–BAFKU_2_: Stress relaxation results for the longest copolymer, H21–BAFKU_2_, are shown in Figure 8, while the results for the remaining two copolymers can be seen in Appendix A.

At low temperatures in the rubbery plateau (−50 °C in Figure 8), there is a small and fast initial relaxation, followed by practical time-independency of the measured stress (on the time scale of 10 min). The initial small relaxation might be due to the disconnection of easily reversible entanglements. At +15 °C, in the warmer temperature region of the rubbery phase, the initial fast relaxation becomes more intense, especially at higher applied strains but the measured stress seems to approach a non-zero final value (in the mid-term). At 0 °C, in the early nematic melt region, the stress relatively quickly relaxes ca. 85% of its value and subsequently stabilizes on a slowly decreasing course. The less than complete rapid stage of the relaxation seems to be connected with permanent entanglements, which are due to long H21 chains and to the stabilization of the entanglements by residual physical crosslinking (via BAFKU lamellae) in the nematic melt. At 20 °C, in the warmer region of the nematic melt, the stress relaxation is nearly immediate: the measured stress value after the applied deformation runs constantly at 0 Pa.

H11–BAFKU_2_: The shorter elastomer H11–BAFKU_2_ displays very similar trends like H21–BAFKU_2_ (see results in Appendix A) but the entanglements’ effect is nearly absent, so that ca. 100% of the stress relaxes in the rapid relaxation stage already at the end of the temperature region of rubber melting (at +5 °C). A new interesting (albeit small) effect in this copolymer is, that at +5 °C and at high applied initial deformation, the stress relaxation in the viscous nematic melting material overshoots the value of zero stress and reaches small negative values. This likely is caused by shear-induced ordering of the nematic lamellae and of the whole lamellar structure, which is dynamic at this temperature. A similar effect of higher strength was observed for H03–BAFKU_2_, which contains more of the mesogen, while the effect is absent in H21–BAFKU_2_.

H03–BAFKU_2_: The melt region of the vitrimer H03–BAFKU_2_ (see Appendix A) displays similar characteristics like the H11–BAFKU_2_ melt but the relaxation of the relatively low-molecular H03–BAFKU_2_ is faster: At +15 °C (zone of glass melting), the fast stage of the relaxation already is nearly quantitative. The ‘relaxation overshoot effect’ is markedly more intense in H03–BAFKU_2_ than in H11–BAFKU_2_ and it occurs even in case of small initial deformations. It is the strongest in the early smectic melt region (at 25 °C).

### 3.5. High-Frequency Stiffening and Self-Healing Effects

Frequency sweep tests (1 mHz to 100 Hz) were performed with the prepared copolymers, in order to evaluate high-frequency stiffening and eventual frequency-induced transitions between liquid-like and rubber-like behavior at several characteristic temperatures. With the elastomers H11–BAFKU_2_ and H21–BAFKU_2_, the tests were conducted in the rubbery and in the liquid state (see Figure 9, as well as Appendix A), whereas with the vitrimer H03–BAFKU_2_, this characterization was performed only in the melt range (Appendix A). The tests proved frequency stiffening in the copolymers, as well as self-healing effects at favorable temperatures, with crystalline as well as with smectic BAFKU lamellae.

An important experimental detail was the division of the frequency sweeps into decades of points for each frequency order, where different deformation amplitudes (higher strains at low frequencies) were applied in order to achieve optimal signal quality. For technical reasons, the rheometer made brief (in the range of seconds) delays between the decades. These delays made possible to observe eventual internal self-healing after endured strain-damage, as steps in the measured curves.

H11–BAFKU_2_: All the three copolymers display a very similar behavior in the frequency sweep tests. As an exemplary system, the results for the copolymer H11–BAFKU_2_ are shown in Figure 9 (selected graphs only, all results are in Appendix A). The following trends can be observed: At low temperatures deep in the rubbery region (e.g., at −60 °C, see Figure 9), *G*′ always prevails over *G*″, even at the lowest frequency of 1 mHz and the samples hence are truly rubber-like. The physical crosslinks are robust in this temperature region, so that the test at −60 °C with high applied strains results in macroscopic mechanical damage (irregular curves, similarly like in the strain sweep test of the same copolymer at −50 °C shown in Figure 5; the damage worsens during the delays). At higher temperatures in the rubbery region (−20 °C), no macroscopic destruction is observed but only shear-thinning damage to the crosslink density. During the experimental delays, the internal damage is at least partly repaired by crosslink re-combination. The self-healing is very fast, as it generates marked effects during the brief (multi-second) delays. As the rubbery phase is fully molten (see example in Figure 9: 10 °C), the shear-thinning at low frequency (combined with large deformation amplitude), as well as the self-healing effects completely disappear and the measured curves become nearly linear in the nematic melt. Already below the melting region, *G*″ starts to dominate over *G*′ but *G*′ increases more steeply with frequency. In the nematic melt and at higher temperatures, *G*′ and *G*″ have a parallel course.

Except in the region of mechanical damage, all three copolymers display frequency-stiffening in the solid and in the molten state, like typical linear polymers.

H21–BAFKU_2_: This sample displays generally nearly identical trends like H11–BAFKU_2_, except for a lower melting temperature (see Appendix A).

H03–BAFKU_2_: The shortest copolymer, H03–BAFKU_2_ (Appendix A), which lacks a rubbery phase, also displays similar trends like H11–BAFKU_2_ and H21–BAFKU_2_, if the viscous melt ranges are compared. An interesting feature are the shear thinning and self-healing effects, which are observed in H03–BAFKU_2_, although it is in the molten state: These effects are limited to the range of the smectic BAFKU aggregates, which split and self-heal similarly like the crystalline aggregates in the ‘warmer rubbery phase’ of the longer copolymers. In the nematic region of the melt, H03–BAFKU_2_ behaves analogically to the longer copolymers.

### 3.6. Thixotropy

The physical nature itself of the crosslinking in the studied copolymers suggests that thixotropy (shear-thinning) effects could be prominent in the molten state and even at the fringe of the rubbery state, if the temperature is not very distant from the respective melting point. The final part of this study hence was dedicated to the evaluation of thixotropy. The effect was evaluated in oscillatory experiments, as well as in steady flow experiments (thixotropic loop). Very strong thixotropy could be documented and also a very fast recovery of the original viscoelastic properties. The latter was nearly instantaneous in case of samples which underwent oscillatory shear and somewhat slower in samples which have endured continuous shear flow.

As first experimental indication, several of the further-above-discussed characterization methods already indicated thixotropy, for example, the multi-frequency gelation tests (Figure 1 and Figure 2, effect of switching the strain amplitude) or the kinetics of physical crosslinking upon melt quenching (Appendix A at 10 °C: decrease of moduli after initial build-up). The mechanical crosslink disconnection observed in the strain sweep experiments (Figure 5 and Figure 6) also can be considered a thixotropic effect. The strong effect of the applied different deformation amplitudes on the measured moduli is well visible in Appendix A.

#### 3.6.1. Oscillatory Tests of Thixotropy and of Recovery of Viscoelasticity

Direct tests of thixotropy and of the recovery of viscoelastic properties are shown in Figure 10, on the example of H11–BAFKU_2_: The sample was subjected to an oscillatory shear deformation with the constant frequency of 1 Hz, while the deformation amplitude was multi-step-wise changed between the values of 0.1% and 5030%.

The tests were conducted at two temperatures: at +20 °C, not far above the melting point and at 45 °C, close to the point of isotropization of the nematic melt. It can be seen that at both temperatures, the storage (elasticity) modulus increases by more than two orders if going from the very high deformation amplitude of ca. 5000% to the small one of ca. 0.1%. The loss modulus displays a markedly lower sensitivity to deformation amplitude than the storage modulus.

Most importantly, it can be noted, that the thinning or the thickening of the copolymer melt occurs practically immediately upon the change of the deformation amplitude, also in case of smaller step-wise changes.

#### 3.6.2. Thixotropic Loop Tests

The thixotropy of the studied copolymers was more profoundly investigated by standard thixotropic loop tests. In the first stage of these tests, the shear rate was continuously raised from 0 to 100 s^−1^ and subsequently continuously reduced again to 0 s^−1^. In a second stage, a fast scan was done of the higher shear rates, where several points were measured, namely at 100, 200, 300, 400, 300, 200 and 100 s^−1^, in this order. In case of sufficiently slow re-generation of the elastic structures in the liquids, a distinct hysteresis would be expected between the ‘shear-rate-up-curve’ and the ‘shear-rate-down-curve’. It was found that the copolymers display mutually similar trends in their respective characteristic temperature regions. The results indicate very fast recovery of flow-induced damage to crosslinking.

H11–BAFKU_2_: The shorter elastomer H11–BAFKU_2_ (see Figure 11) generates a considerable resistance, if measured in the boundary region between rubber and the liquid nematic phase at +5 °C. Only a limited shear rate range could be investigated, up to 30 s^−1^. A visible albeit not very large hysteresis of the curves of shear-rate-dependent stress and viscosity is observed at this experiment temperature, as well as oscillations in the ‘down-curve’ at lower shear rates. Both effects likely can be attributed to the behavior of the lamellar BAFKU nano-aggregates, which are on the boundary between the frozen crystalline state and the more dynamic nematic state. Their re-organization upon shearing is no more very fast, so that some shear damage persists on the time scale of the experiment. The oscillations can be possibly caused by the tumbling of fragments of the lamellar structure, in which the BAFKU lamellae are crystal-like and rigid. At +20 °C, in the middle of the nematic temperature region, the hysteresis area between the thixotropic curves is very slim, both in the continuous loop up to 100 s^−1^, as well as in the test of the high shear rates (up to 400 s^−1^). At a first glance, the nearly negligible hysteresis might appear as a contrast to Figure 10 further above, where at the same temperature, oscillatory shear of high amplitude generates a thixotropic drop in modulus by nearly three orders. However, in Figure 10, the modulus (and hence the stress, as well as the complex viscosity) also is shown to recover immediately (in less than 1 s) if the oscillatory amplitude is reduced. This ultra-fast recovery in fact is in good agreement with the very slim hysteresis observed in the loop experiments in Figure 11. The still observed small hysteresis means that in continuous flow, some more intense damage to the elastic structure is done, which is not fully and immediately recovered during the continuous change of the shear rate. If the shear-rate-induced decrease in viscosity is evaluated, it can be noted that most of the shear thinning occurs between 0 and 20 s^−1^ (largest part of it between 0 and 5 s^−1^).

At 80 °C, in the isotropic melt region, the hysteresis in the thixotropic loop graph completely disappears, while an initial drop of viscosity upon the onset of shearing is observed, similarly like at the lower temperatures but it occurs in a narrower range of shear rates. After that, the characteristics are Newtonian. The initial high viscosity can be attributed to ‘soft crosslinking’ by droplet-like isotropic aggregates of BAFKU end-groups

H21–BAFKU_2_: The longest copolymer H21–BAFKU_2_ (see Appendix A) displays similar trends in thixotropy loops like H11–BAFKU_2_. In spite of the lower melting point, its melt at 0 °C generates a considerable resistance (and elasticity), as noticed already in case of creep experiments. Standard thixotropic loop tests were possible only with small maximum shear rates, where very high viscosity and a very marked hysteresis (Appendix A, top row) were observed. Due to the low content of the physically crosslinking BAFKU units, however, this copolymer can be more easily liquefied by intense shearing, than the shorter ones. After forcefully applying a 100–400–100 s^−1^ loop, a dramatic shear thinning was achieved. A subsequent measurement of the continuous thixotropic loop (0–100–0 s^−1^) yields a wholly different thixotropic profile than in case of an intact sample (see Appendix A, last graph of top row), characterized by small viscosity (nearly two orders below original values) and a slim hysteresis. The liquefication is possible because the recovery of elastic structures is slow at 0 °C in H21–BAFKU_2_. At 20 °C, in the nematic region, the hysteresis of H21–BAFKU_2_ is very slim but discernible in the continuous loop up to 100 s^−1^ and well visible albeit still moderate in the loop up to 400 s^−1^. This stronger tendency to hysteresis might be due to the mentioned entanglements. At 80 °C, H21–BAFKU_2_ displays no more hysteresis, in analogy to H11–BAFKU_2_.

H03–BAFKU_2_: The shortest copolymer, H03–BAFKU_2_ (see results in Appendix A), displays the strongest hysteresis tendency in its thixotropic loop characteristics among the studied products. Only in the isotropic melt at 80 °C, (see Appendix A), the hysteresis is nearly absent. The nematic phase of H03–BAFKU_2_ (50 °C, see Appendix A) displays the most distinct hysteresis among the nematic phases of the studied copolymers, as well as irregular oscillations of the curves. The hysteresis is even wider in the smectic phase (35 °C, see Appendix A), where also the strongest oscillations of the curves are observed, if shear rates higher then ca. 2 s^−1^ are applied. At 25 °C in the smectic phase, only limited shear rate scans were possible, which also display a significant hysteresis. Generally, it seems that in H03–BAFKU_2_, except at isotropic temperatures, the splitting of lamellae, as well as tumbling of fragments of lamellar structure (somewhat similar to domain tumbling in LC polymers discussed in Reference [37]) plays a strong role in the flow behavior and in thixotropic loop tests.

### 3.7. Comparison of Rheological Behavior of the Studied and of Classical LC Polymers

In view of all of the above-discussed results, it is interesting to compare the mechanisms of flow in the studied copolymers and in classical liquid-crystalline polymers (LCPs). The latter were discussed in the Introduction: The model of three regions of flow [31,32], as well as the Larson-Doi polydomain model [33,34] well describe the rheology of most LCPs, which are typically very rich in mesogen units. The flow mechanism corresponding to these models is summarized in Scheme 10: At the lowest shear rates, a tumbling of small oriented LC domains occurs (region I). At higher shear rates, these domains gradually diminish and an increasing number of macromolecules join the more or less flow-aligned ‘nematic sea’, while some domains still persist (region II). Finally, in region III, all material consists of the ‘nematic sea’. Structural features of the respective LCPs influence the prominence and the properties of the rheological regions.

In contrast to the typical LCPs, the presently studied copolymers are much less mesogen-rich, they contain 52, 37 and 16 Vol.% of mesogen, respectively. Additionally, they undergo strong nano-phase separation of the highly flexible central PDMS chains from the rigid mesogenic diaromatic azo end-groups. This phase separation, which persists also in the liquid phase, appears to play a key role in the mechanical as well as in the rheological properties of the studied copolymers. Scheme 11 illustrates the main factors, which appear to play a role in the flow behavior of the presently studied copolymers: (1) the splitting and recombination of the lamellae formed as result of the phase separation, which are liquid-crystalline in the molten polymers at not overly high temperatures; (2) the thermotropic phase transitions in these lamellae, which change their stiffness and the dynamics of their splitting and re-combination; both the effects (1) and (2) seem to be responsible for the multiple gel points observed in the copolymers; (3) finally, also the entanglements of the elastic PDMS chains were found to play a certain role.

If comparing the presently studied PDMS–BAFKU_2_ compounds with a copolymer of similar basic architecture studied by Winter, Lin and co-workers [52], some similarities can be noted, like a higher viscosity of the nematic melt than of the isotropic, as well as the possibility of shear induced transition from solid to liquid state. Both effects in both systems are connected with the nano-phase-separation of the LC end-groups. The comparison further shows, that in the presently studied copolymers, the much-increased flexibility of the central chain, as well as a higher phase-separation tendency of the constituent segments, tremendously increase the rate of recovery of strain- or even of flow-induced damage (e.g., the above-discussed immediate recovery during oscillatory tests).

## 4. Conclusions

−Physically crosslinked thermo-reversible low-temperature-melting rubbers, based on linear polydimethylsiloxane (PDMS) capped in α,ω-positions with liquid-crystalline (LC) building blocks called “BAFKU”, were studied concerning their viscoelastic and rheological properties in melt, as well as in the rubbery state.−The properties of three copolymers were compared, which differed in the length of the central PDMS chain, namely DMS H03 (8.6 dimethyl siloxane repeat units), DMS H11 (16.3-mer) and DMS H21 (64.4-mer).−Physical crosslinking via nano-aggregation of BAFKU units in all three tested copolymers was found to be fairly efficient and very large step-wise changes of elasticity and viscosity were observed, which correlated with the thermotropic properties of the crosslinks.−The rheological behavior of the copolymers was found to be controlled by an interplay of nano-phase separation of the LC end-groups (growth and splitting of their aggregates) and of the thermotropic transitions in these aggregates (which change their stiffness). Entanglements of the elastic PDMS chains also were found to play a role.−In contrast to LC-rich liquid crystalline (co)polymers (LCPs), the studied copolymers display viscosity increase if going form isotropic to nematic (or further to the smectic) state, because their viscoelasticity is controlled by the larger-scale morphology, which is responsible for the physical crosslinking. The latter in turn is controlled by nano-phase separation and by the strengthening or loosening of the aggregates of LC units via thermotropic transitions.−The copolymers display up to three gel points, if a temperature scan (in both directions) is performed in the range from rubbery state to isotropic melt. The gel points correlate with the thermotropic transitions (I/N, N/Sm and Sm/Cr in the shortest copolymer and I/N, N/Cr in the longer ones) and with the associated changes in stiffness and dynamic size of LC nano-aggregates.−The kinetics of (physical) gelation to network is fairly fast if induced by temperature (abrupt melt cooling): between 0.5 and 3 min (typically ca. 1 min). It slows down at higher final temperatures. This kinetics is controlled by temperature-dependent nano-phase-separation dynamics.−The physical crosslinks can be reversibly disconnected by large mechanical strain in the rubbery state and in the melt (thixotropy in the latter case). The kinetics of subsequent re-generation was found to be extremely fast: ca. 1 s in oscillatory tests.−Thixotropic loop tests, in which the samples were subjected to continuous flow, also indicate a very fast regeneration of destroyed physical crosslinks, so that only very small hysteresis is observed in these tests, in spite of very strong shear-thinning tendency in all the copolymers in wide temperature ranges. Nevertheless, in contrast to oscillation experiments, the crosslink regeneration in continuous flow is not immediate.−The ‘warmer’ rubbery phase of the longer copolymers can be relatively easily transformed to liquid by strong shear.−Frequency-stiffening was observed in the rubbery state, as well as in the melt. Such a behavior is characteristic of classical elastomers and linear polymers.−The studied low-temperature elastomers might be of interest as passive smart materials for advanced applications such as viscoelastic coupling for example, in soft robotics (transitions melt/viscoelastic melt/rubber) but also as damping materials (energy absorption via physical crosslink disconnection). Additionally, the incorporated LC building blocks of azo type open the possibility of reversible UV-light-induced switching of material properties.

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
