# Peer review of "Low-Temperature-Meltable Elastomers Based on Linear Polydimethylsiloxane Chains Alpha, Omega-Terminated with Mesogenic Groups as Physical Crosslinker: A Passive Smart Material with Potential as Viscoelastic Coupling. Part II—Viscoelastic and Rheological Properties"

_polymers, 2020, doi:10.3390/polym12122840_

Round 1
Reviewer 1 Report
The manuscript contains a thorough rheological and viscoelastic characterization of linear polydimethyl-siloxane of different chain lengths end capped by aliphatic chain spacers and mesogenic azobenzene groups. It is a systematic and very detailed study though a wide range of temperatures covering all phase transitions. I recommend publication after rechecking and elimination of some syntax errors and resolving the few minor issues listed below
In the Schemes 3-5 the authors should explain the use of different colors at the temperatures. For example I assume that the transition temperatures at the top refer to heating (red) and cooling (blue) runs.
I have some reservations concerning Scheme 6. Do the authors have proof that the rigid moieties are perpendicular to the direction of the layers in the smectic state? In addition the “molten” dimethyl siloxane parts in (b) must be depicted somewhat differently from their solid analogues (a).
I also object to the use of the copolymers since the BAFKU part is not a polymer. I would suggest the terms end capped polymers, composite polymers or hybrid flexible-rigid rod polymer instead.
Page 24 Lines 27-29: “It can be seen, that in the low-temperature part of the rubbery region (-50°C in Figure 7) nearly no creep (in view of the error margin) occurs on the time scale of 3 min at any applied stress (up to 10 000 Pa).” The corresponding diagram is not in Figure 7. It is of course quoted in the supporting information section and I strongly suggest its inclusion to the main manuscript.
Reviewer 2 Report
The paper presents very well conducted and comprehensive analysis of rheological properties of the newly designed end-chain functionalized elastomers. The scientific soundness and value of this article is very high. I have only a two minor points regarding this article and I recommend it to publication.
- The document is not edited accordingly to the journal template
- in the Scheme 1 there is a mistake regarding to the length of the PDMS unit: 64.2 (total: 64.4) Me2SiO units - should be - 62.4 (total: 64.4) Me2SiO units
This manuscript is a resubmission of an earlier submission. The following is a list of the peer review reports and author responses from that submission.
Round 1
Reviewer 1 Report
My recommendation is to reject this contribution in its current state. The work presented in this contribution and the preceding is from my point of view solid and it could be partitioned into two manuscripts, probably in a different manner. However the authors fail to provide this second contribution with its own significancy, which is the characterisation of the viscoelastic properties of the materials. I will mention just several facts:
1.- First half of the abstract is identical to the first half of the abstract of the first paper (Part I). Beyond other considerations it is too much to be dedicated to work presented in a different contribution.
2.- The introduction is nearly identical to that of previous paper (Part I). Part I is about design, synthesis, chemical characterization and phase behaviour of the materials and the introduction is suitable to that topic. The central focus of this contribution is on rheology and as such I suggest that the introduction could center on rheology of similar materials, advantages of low temperature elastomers, or even on the rheology of LC phases.
3.- The authors over cite their own work (if we just take the corresponding author, 20 citations out of 57)
4.- Scheme 1 -4 are copied-pasted from Part I.
5.- Figure 1 presents the same results as Figure 1 of Part I
6.- Pages 5-9 comment results of the other contribution.
7.- First 3 conclusions are actually conclusions of Part I.
Reviewer 2 Report
I suggest to rewrite completely the paper, since in this form it seems more the extract of a thesis and it has not the format and the style of a scientific paper. First, I suggest to select the relevant results and focusing on those parts. For instance, some figures (Figures 2 and 3) are not readable and very difficult to understand.